# Ethnic and social inequalities in COVID-19 outcomes in Scotland: protocol for early pandemic evaluation and enhanced surveillance of COVID-19 (EAVE II)

Paul Henery [1,2] Eleftheria Vasileiou [3] Kirsten J Hainey [1]
Duncan Buchanan,[2] Ewen Harrison,[3] Alastair H Leyland,[1] Thomas Alexis,[4]
Chris Robertson,[5] Utkarsh Agrawal,[6] Lewis Ritchie,[7] Sarah Jane Stock [2,8]
Colin McCowan,[9] Annemarie Docherty,[10] Steven Kerr,[3] James Marple,[11]
Rachael Wood [3,12] Emily Moore,[2] Colin R Simpson,[3,13] Aziz Sheikh,[3]
Srinivasa Vittal Katikireddi [1,2]

For numbered affiliations see end of article.

**Correspondence to**
Dr Srinivasa Vittal Katikireddi;
vittal.katikireddi@glasgow.ac.uk

## ABSTRACT

**Introduction** Evidence from previous pandemics, and the current COVID-19 pandemic, has found that risk of infection/severity of disease is disproportionately higher for ethnic minority groups, and those in lower socioeconomic positions. It is imperative that interventions to prevent the spread of COVID-19 are targeted towards high-risk populations. We will investigate the associations between social characteristics (such as ethnicity, occupation and socioeconomic position) and COVID-19 outcomes and the extent to which characteristics/risk factors might explain observed relationships in Scotland.

The primary objective of this study is to describe the epidemiology of COVID-19 by social factors. Secondary objectives are to (1) examine receipt of treatment and prevention of COVID-19 by social factors; (2) quantify ethnic/social differences in adverse COVID-19 outcomes; (3) explore potential mediators of relationships between social factors and SARS-CoV-2 infection/COVID-19 prognosis; (4) examine whether occupational COVID-19 differences differ by other social factors and (5) assess quality of ethnicity coding within National Health Service datasets.

**Methods and analysis** We will use a national cohort comprising the adult population of Scotland who completed the 2011 Census and were living in Scotland on 31 March 2020 (~4.3 million people). Census data will be linked to the Early Assessment of Vaccine and Anti-Viral Effectiveness II cohort consisting of primary/secondary care, laboratory data and death records. Sensitivity/specificity and positive/negative predictive values will be used to assess coding quality of ethnicity. Descriptive statistics will be used to examine differences in treatment and prevention of COVID-19. Poisson/Cox regression analyses and mediation techniques will examine ethnic and social differences, and drivers of inequalities in COVID-19. Effect modification (on additive and multiplicative scales) between key variables (such as ethnicity and occupation) will be assessed.

### Strengths and limitations of this study

► This study uses a near-complete cohort of the Scottish population, providing considerable statistical power to investigate social inequalities in COVID-19.

► Linkage of the Scottish Census to administrative data allows for evaluation of differences in COVID-19 across specific occupational classes, which is not available in any other dataset in Scotland.

► Given the Census data are from 2011 and are being linked to data from 2020 to 2021, some occupational and social data may be out of date.

► Individuals who moved to Scotland post-Census will not be included in all analyses, for the same reasons as above.

**Ethics and dissemination** Ethical approval was obtained from the National Research Ethics Committee, South East Scotland 02. We will present findings of this study at international conferences, in peer-reviewed journals and to policy-makers.

## INTRODUCTION

Historically, pandemics have disproportionately affected specific groups within the population. For example, during the Spanish influenza pandemic of 1918, individuals from poorer socioeconomic positions (SEP) experienced higher incidence of influenza in the USA.[1] These findings have been replicated in various international settings using data from the 1918 pandemic.[2] Similar patterns were observed following the H1N1 influenza pandemic in 2009 in Canada, with an increased risk of hospitalisation among

individuals with lower educational attainment or resident in areas of greater deprivation.[3] English data demonstrated a socioeconomic gradient in mortality during the 2009 pandemic, with the highest risk of death observed in those living in deprived areas.[4 5] Ethnicity was also identified as a risk factor for mortality during the 2009 pandemic, with those in non-White ethnic groups at increased risk of mortality from H1N1 influenza compared with White populations in the UK.[5] The mechanisms underlying these socio-economic patterns have been hypothesised as resulting from a combination of different exposure and susceptibility to infection, and differing access to healthcare services.[6] Understanding the role of ethnic, socioeconomic and occupational risk factors in both risk and outcome of infection during a pandemic is essential for health service planning, targeting preventative measures and informing future modelling efforts.

The SARS-CoV-2 and its resulting disease (COVID-19) is spreading rapidly worldwide, causing a high burden of mortality and morbidity.[7] The differential risks of infection in ethnic minority groups have been examined during the COVID-19 pandemic.[8] Epidemiological analyses within England have found higher risks among some minority ethnic groups, both for risk of infection and for the consequences of infection.[9–13] These findings have been supported by international evidence, which suggests individuals from Black and South Asian ethnic groups are at increased risk of experiencing COVID-19 and its resultant harms.[12 14]

Emerging evidence suggests that marked socioeconomic inequalities in risk and outcome of COVID-19 exist, yet empirical evidence describing the role of socioeconomic factors in Scotland is limited.[11] Initial analyses in Scotland have been subject to considerable misclassification of ethnicity (due to systematically incomplete data within health records) and limited by using a broad three-category classification of ethnicity in a small case–control sample[15]; however, survey-based research with more specific ethnicity classifications has suggested that ethnic minorities may be more hesitant to be vaccinated against COVID-19.[16] Analyses have therefore shown inconsistent findings. Understanding how sociodemographic factors are related to infection risk and prognosis, as well as monitoring inequalities by area level (such as indices of multiple deprivations) and individual/household-level (such as social class or household tenure) SEP, is essential for health service planning, targeting prevention efforts and informing future modelling efforts. In addition, our understanding of why inequalities in COVID-19 outcomes exist remain limited.[17] For example, differences in occupation, household composition and healthcare could operate as important mediating mechanisms, which could potentially be targeted to mitigate ethnic inequalities in COVID-19.

The Early Assessment of Vaccine and Anti-Viral Effectiveness II (EAVE II) study[18–20] is an ongoing project which tracks the epidemiology of COVID-19 in Scotland in real time, and aims to evaluate the effectiveness of future vaccinations in reducing spread of the infection. While the main EAVE II project will stratify COVID-19 rates and vaccine effectiveness by demographic characteristics such as age and region, it will not examine the role of ethnicity and social factors in detail. Linkage of the EAVE II dataset to other administrative sets, in particular the 2011 Scotland Census, allows the assessment of the role of social and ethnic inequalities on risk of SARS-CoV-2 infection and subsequent disease outcomes. This provides better understanding of groups at higher risk from COVID-19 within the Scottish population compared with administrative data sources (which typically lack detailed information about social characteristics), yielding valuable information for clinicians and healthcare planners as the pandemic continues to evolve. Learning from this pandemic may also yield longer term lessons for public health policy.

## Aim and objectives

This EAVE II substudy aims to quantify social inequalities in COVID-19 related outcomes in Scotland using linked census and administrative data, as well as examining the mechanisms through which inequalities in COVID-19 arise.

Our primary objective is to describe the epidemiology of COVID-19 by: (1) ethnicity, (2) SEP, (3) occupation and (4) residential circumstances.

In addition, we have five secondary objectives:

1. To monitor receipt of COVID-related treatment and preventive interventions across social and ethnic groups.
2. To quantify ethnic and social differences in the risk of adverse outcomes (hospital admission, critical care admission and mortality) from COVID-19.
3. To explore the potential contribution of pre-existing health conditions and risk factors to ethnic and social inequalities in COVID-19 and prognosis.
4. To investigate if occupational (and potentially socio-economic or ethnic) differences in COVID-19 risks are modified by other social factors.
5. To assess the quality of ethnicity coding within National Health Service (NHS) administrative datasets.

Secondary objective b will be examined for three groups defined by experiencing: (1) laboratory confirmed SARS-CoV-2 infection, (2) clinically diagnosed SARS-CoV-2 infection and (3) serological evidence of previous infection with SARS-CoV-2.

## METHODS AND ANALYSIS
### Study design and population

We will perform analyses on a national prospective cohort of the Scottish population, including all individuals who meet the following criteria: (1) alive on the 1 March 2020; (2) resident in Scotland on that date and (3) present in the 2011 Census in Scotland. A start date of 1 March 2020 has been selected as the first case of laboratory confirmed SARS-CoV-2 in Scotland was identified on 2 March 2020.[21]

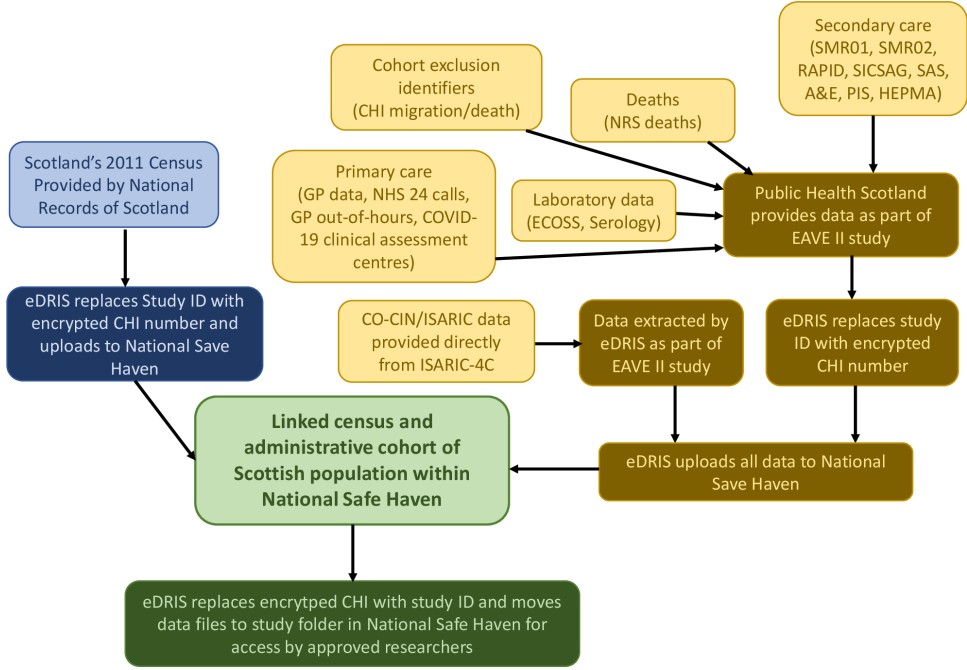

**Figure 1** Data flow diagram for linkage of census and administrative data comprising cohort blue box: census data yellow box: administrative data green box: linked data dark background: data processing. A&E, Accident and Emergency; CHI, Community Health Index; CO-CIN, COVID-19 Clinical Information Network; EAVE II, Early Assessment of Vaccine and Anti-Viral Effectiveness II; ECOSS, Electronic Communication of Surveillance in Scotland; eDRIS, electronic Data Research Innovation Service; GP, general practice; HEPMA, Hospital Electronic Prescribing and Medicines Administration; ISARIC, International Severe Acute Respiratory and Emerging Infection Consortium; NHS, National Health Service; NRS, National Records for Scotland; PIS, Prescribing Information System; RAPID, Rapid Preliminary Inpatient Data; SAS, Scottish Ambulance Service; SICSAG; Scottish Intensive Care Society Audit Group; SMR, Scottish Morbidity Records.

Analyses will focus on the adult population aged 16 years and over, with some analyses limited to older age groups (eg, aged 25+ years on the Census 2011 date when analysing highest education level). We will also conduct analyses that include individuals not present in the 2011 Census when they are of specific policy interest—for example, to look at vaccination uptake.

### Databases

The Community Health Index (CHI) is a unique numerical identifier, assigned to every individual registered to receive healthcare in Scotland, which can be used to identify and link all healthcare interactions and records for an individual.[22] Linkage of these administrative datasets with the 2011 Census data will allow the examination of clinically diagnosed and laboratory confirmed SARS-CoV-2 infection, deaths and potential confounders (figure 1). The CHI is mandatory for anyone receiving medical treatment.[22] Furthermore, individuals known to have emigrated (through general practice (GP) deregistration) prior to the pandemic will be excluded. Data linkage and analyses will take place within the National Safe Haven, a secure and trusted research environment.[23]

### Census

In Scotland, the national Census is conducted every 10 years by means of a mandatory survey sent to every household in the country.[24] The most recent Census was conducted in 2011 and forms the population spine of our cohort. The census extract for this study contains demographic information including ethnic group, religion and country of birth, marital status, SEP indicators such as highest educational attainment, employment status and occupation, household information including persons per room, tenure and number of cars and variables on prepandemic health status identifying long-term illnesses and disability status. Census data are necessary to identify our exposures of interest, as well as to study potential mediating pathways and effect modification. The necessary information required from the Census (eg, ethnicity, occupation) is not reliably available from other sources. While ethnicity is recorded in some health datasets in Scotland, it is often missing, reported inconsistently across datasets, and not always based on self-report, which is considered the gold standard for establishing ethnicity.[25]

### Laboratory data

To identify individuals with confirmed SARS-CoV-2 infection, positive viral reverse transcriptase PCR (RT-PCR) results will be confirmed using Electronic Communication of Surveillance in Scotland (ECOSS) data, a national repository of diagnostic and reference laboratory data on testing for viral infections including COVID-19.[26] To identify individuals with serologically confirmed infection,

serology data will be provided via the Enhanced Surveillance of COVID-19 in Scotland (ESoCiS) programme undertaken by Public Health Scotland (PHS) on behalf of the Scottish Government.[27]

## Secondary care

The Scottish Morbidity Record (SMR), maintained by PHS, is the primary repository for data describing hospital admissions in Scotland.[28] We will use data from SMR01, which details general inpatient admissions, and SMR02, which details obstetric admissions. We will include data from the Rapid Preliminary Inpatient Data (RAPID) dataset. The RAPID data are collected quickly, in an unprocessed state and are therefore available more quickly than SMR data,[29] which is necessary given the pertinent nature of our study. Data about intensive care unit admissions in adults will be included by linking data from the Scottish Intensive Care Society Audit Group (SICSAG).[30] These datasets use International Classification of Diseases (ICD-10) codes to record an individual's diagnoses. ICD-10 codes will be used to identify a clinical diagnosis of COVID-19 in the absence of a positive test result for those admitted to hospital with diagnosed COVID-19. Scottish Ambulance Service (SAS) and Accident and Emergency (A&E) records, also provided by PHS, will be used to identify people with clinically suspected COVID-19.

The prescribing information system (PIS) is the repository for all prescriptions requested for individuals by general practices,[31] and the Hospital Electronic Prescribing and Medicines Administration (HEPMA) is the equivalent for prescribing and administration of medicine in a hospital setting.[32] Both of these datasets use the British National Formulary (BNF), a standardised medication classification system.[33] We will use BNF codes to identify prescribed drugs or administered medicine used to treat COVID-19, identify comorbidities (such as diabetes and hypertension) and allow the study of longer-term consequences of COVID-19.

## Primary care

Just under 100% of the population of Scotland is registered with a GP, which provides free-at-point-of-delivery primary healthcare services.[34] NHS 24 is the Scottish national telephone service for the general public, which provides clinical triage and advice for those experiencing non-critical ill health during the out-of-hours period. This was expanded to provide advice to individuals with suspected COVID-19 at any time during the pandemic.[35] We will use GP consultation data and NHS24 out-of-hours data to identify clinically suspected COVID-19. In addition, data from the recently established COVID-19 clinical assessment centres will also be included for this purpose.

## Deaths

National Records for Scotland (NRS) provides data about deaths attributed to COVID-19, which will be included in our dataset.[36] ICD-10 codes will be used to identify

deaths occurring in individuals with clinically confirmed COVID-19 within 28 days of death (U07.1, U07.2).

## Exposures and covariates

We will examine four categories of social factors in relation to the study's primary outcomes: ethnicity, SEP, occupation and residential circumstances. Ethnicity will be categorised based on standard Census classifications,[37] self-reported in sixteen groups. We will try to analyse ethnicity using as detailed categorisations as possible, subject to having adequate numbers for meaningful statistical analysis. SEP will be classified at both area level (the Scottish Index of Multiple Deprivation (SIMD)) and individual level (eg, education level and housing tenure). The SIMD is a measure of the relative deprivation of broadly heterogeneous population level 'data zones' within Scotland, classifying deprivation using the following weighted domains: employment, income, health, crime, housing, education and access to services.[38] The SIMD is typically operationalised using either quintiles or deciles, with quintile/decile 1 referring to the most deprived 20% or 10% of the population, respectively. Marital status will consist of five categories as per the Census: single (never married), married, separated, divorced and widowed.[39] Housing tenure will consist of seven categories as per the Census: owned (outright), owned (mortgage), social rented (council), social rented (other), private rented (landlord/letting agency), private rented (other/rent-free) or communal.[40] Occupation will be measured using the Standard Occupational Classification 2000, consisting of major, submajor, minor and unit group levels.[41] Employment status, weekly hours worked and type of industry (coded using Standardised Industrial Classification, SIC) will be used to explore potential differential workplace exposure. Residential circumstances will be classified using the following parameters: housing tenure, persons per room, household size and age range of household (difference between oldest and youngest permanent occupant).

The following additional characteristics will be used to address confounding, explore mediation and for assessing effect modification. Cultural factors will be investigated, including religion, and English language ability via a four-point Likert scale (very well, well, not well, not at all). Geographical variables will include 32 local authorities (LA) comprising local government in Scotland. While these LA have some devolved powers, primarily in service provision,[42] the most pertinent local factor for this study is that COVID-19 mitigation levels are, at the time of writing, allocated by a LA-level tiering system.[43] We will explore the role of rurality using the Scottish Government Urban Rural Classification, which can be used as a simple two-fold measure (defining 'rurality' as areas with a population of <3000) at its simplest, or an eight-fold measure at its most complex.[44] Prepandemic health status will be assessed on the basis of general self-reported health, captured using a five-point Likert scale (very good, good, fair, bad, very bad), and the presence of the

following additional self-reported health circumstances as recorded in the Census: long-term condition, mental health condition, other condition, hearing impairment, visually impairment, learning disability, developmental disorder, and physical disability. Behavioural risk factors (such as smoking status and body mass index) and health conditions (including pregnancy) will also be identified from GP and hospital data.

## Outcome definitions

The primary outcomes of this study will be laboratory confirmed SARS-CoV-2 infection, clinically diagnosed COVID-19 and serologically confirmed evidence of previous SARS-CoV-2 infection. Laboratory confirmed SARS-CoV-2 infection will be defined as positive RT-PCR via the ECOSS dataset. Serologically confirmed SARS-CoV-2 infection will be defined as positive antibodies on serology testing as identified through ESoCiS data, reflecting immunological evidence of previous infection. Clinically diagnosed COVID-19 disease will be defined as the requisite ICD-10 code being recorded on discharge from hospital via SMR, intensive care via SICSAG, or on NRS death records. Clinically suspected COVID-19 disease will be used as a secondary outcome in the absence of laboratory confirmed COVID-19 and recorded via SAS or A&E attendance records, GP consultation or out-of-hours records, COVID-19 clinical assessment centre data, or NHS 24 logs. Given testing practices are changing over time, we will conduct sensitivity analyses by stratifying analyses by time period and consider weighting analyses to account for differential testing practices.

Secondary outcomes include uptake of vaccination for COVID-19, as recorded in GP records or the vaccine delivery app, and being prescribed treatment for COVID-19 either via the PIS or HEPMA. We will also examine adverse outcomes following COVID-19 infection: admission to hospital, critical care admission or death. Hospital admission or death will be defined on the basis of either COVID-19 (laboratory confirmed or clinically diagnosed) recorded on a hospital admission in SMR01 or death record (using ICD10 codes U07.1, U07.2,) or positive test, or admission in RAPID within 14 days of a SARS-CoV-2 RT-PCR positive test.[45] Critical care admissions will be defined on the basis of presence on the SICSAG database and a relevant hospital admission (using ICD10 codes U07.1, U07.2) or positive test.

## Statistical analysis
### Epidemiology of COVID-19 by ethnicity and social factors
We will describe the epidemiology of COVID-19 by ethnic and social factors, specifically in relation to: (1) incidence of confirmed SARS-CoV-2 infection; (2) incidence of clinically diagnosed disease and (3) evidence of previous infection. We will use summary statistics to describe the cohort by presence and absence of confirmed, diagnosed and suspected COVID-19. We will perform a descriptive analysis of the number of cases by exposure groups in the first instance, to determine the level of disaggregation

feasible. In addition to analysing inequalities by ethnicity and social factors, we will also investigate the inequalities by country of birth (ie, migration-related inequalities).

Poisson regression with robust standard errors will be used to estimate risk ratios for the risk of infection across exposure groups. We will consider age and sex to be key potential confounders for all analyses, with adjustment for LA also planned for most analyses (to account for the differential spread of the infection across geographies, and local mitigation levels). If case numbers allow, we will examine the incidence of confirmed and suspected COVID-19 by sex (and for ethnicity, country of birth). We can assess whether findings are robust to more stringent definitions of confirmed and suspected infection, for example, by restricting the definition of a confirmed case to one with positive viral RT-PCR, and restricting the definition of a suspected case to exclude people with a negative viral RT-PCR result.

### Surveillance of preventative treatment and interventions for COVID-19 across social and ethnic factors
We will describe treatment for and prevention of COVID-19 by ethnicity and other social factors (eg, SEP, occupation), especially in relation to vaccination uptake. We will tabulate those who have been vaccinated as a proportion of the total eligible in different social subgroups, from when the vaccine is available and delivered nationally. This will be monitored over time to investigate whether there is a change in vaccination rates for different subgroups. If we have sufficient numbers of people vaccinated, we will also stratify these analyses by demographic factors. Similarly, we will also investigate the potential ethnic and other social differences in treatment for COVID-19 during acute infection and post-COVID via GP prescribing or hospital-administered medical care.

### Inequalities in risk of adverse health outcomes from COVID-19
We will quantify ethnic and social inequalities in the risk of adverse outcomes (hospital admission, critical care admission and mortality) from COVID-19. We will perform a descriptive analysis of the number of cases by exposure group in the first instance, to help determine the level of disaggregation feasible. We will follow a previously published approach[46] to analyse COVID-19 survival outcomes using Cox proportional hazards models from the date of a positive test result (or first date of diagnosis). We will conduct sensitivity analyses excluding people resident in care homes at the study start date. We will consider Fine and Gray competing risks models (to account for non-COVID-19 deaths) for sensitivity analysis[47] or alternatively by applying inverse probability weights to address competing risks.

### Mediation analyses to explore drivers of inequalities in COVID-19
We will explore the contribution of several potential mediators of ethnic and social inequalities in COVID-19. The extent of these analyses will be determined by the magnitude of any observed inequalities in COVID-19 outcomes.

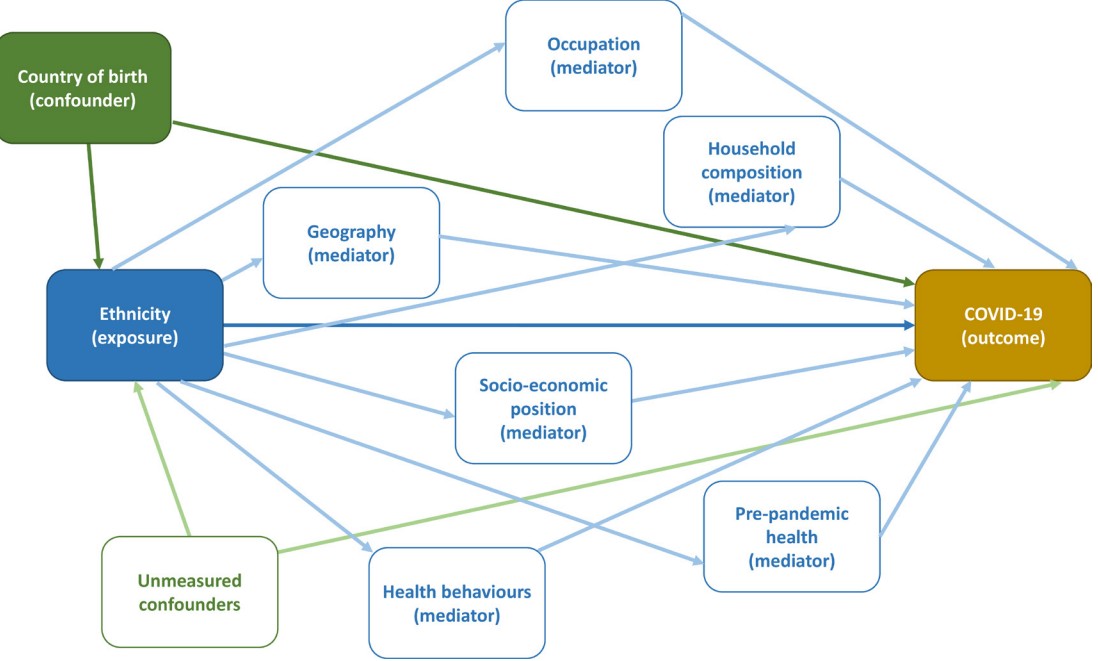

**Figure 2** Directed Acyclic Graph of the causal pathways between ethnicity and COVID-19 yellow box: outcome blue box: exposure green box: confounder white box with green outline: confounder (unmeasured) white box with blue outline: mediator blue arrow: main effect green arrow: confounding light green arrow: unmeasured confounding light blue arrow: mediated effect.

Potential mediators to be investigated for ethnic inequalities include occupation, SEP, cultural factors (such as language ability and country of birth) and prepandemic health (assessed through census, GP records and hospitalisations data). We will conduct cross-tabulations of the relationships between exposures, potential mediators and outcomes, followed by a set of nested regression models for the same purpose. Following this, we will pursue more formal mediation analyses. We expect to use G methods,[48] such as marginal structural modelling, to estimate the direct and indirect effect of ethnicity mediated through the mediating factors. We will investigate the extent to which inequalities might arise from differential exposure compared with differential susceptibility.[49] Our analysis will be informed by a Directed Acyclic Graph (figure 2). Given we are uncertain about whether the country of birth and geography are confounders or mediators, we will conduct sensitivity analyses without adjusting for these variables. In addition to analysing inequalities by ethnicity, we will investigate inequalities by country of birth to explore migration-related inequalities. We will consider conducting similar mediation analyses for socioeconomic inequalities in COVID-19, depending on the extent of inequalities observed.

### Effect modification
We will investigate whether interactions exist for key combinations of exposure variables, such as ethnicity, SEP and occupation. We will study whether occupational risks of COVID-19 differ by ethnic group. Occupation will be assessed on the basis of Standard Occupational Classification 2000 codes, aggregated from three-digit codes to a level that meets disclosure control requirements. We will perform descriptive analysis of the number of cases by occupation initially, followed by cross-tabulation by ethnicity. We will then estimate relative risks for outcomes of interest and check for evidence of effect modification on an additive and multiplicative scale. Should we observe evidence that occupational risk differs by ethnicity, we will calculate relative excess risks due to interaction.[50] Similar analyses will be conducted to assess potential effect modification for key combinations of exposure groups, as prioritised by our policy partners.

### Assessment of quality of ethnicity coding
We will assess the quality of ethnicity coding within NHS administrative datasets, in order to assess whether, in the absence of Census data, administrative data can be reliably used to study ethnic inequalities. We will start by assessing the proportion of information on ethnicity missing within each dataset. Following this, we will consider the 2011 Census to be our 'gold-standard' dataset for comparison purposes. We will quantify the prevalence of different ethnic groups within each dataset and assess the sensitivity, specificity, positive predictive value and negative predictive value, following the methods used in previous validation studies.[51] If numbers of cases allow, we will stratify analyses by broad age group, sex and country of birth.

### Sample size
Accounting for non-linkage and non-participation in the Census (via migration or birth after the Census or otherwise), we estimate that the size of the cohort will be 4.3 million people. We are confident based on research from previous pandemics in similar-sized populations[52]

that there will be sufficient statistical power to perform our planned analyses. For example, for the African ethnic group which is one of the least frequent ethnic groups in Scotland, considering COVID-19 deaths (the least common outcome), we anticipate there being approximately 30 COVID-19 deaths from the first wave in the African ethnic group which exceeds the number of events in previous completed analyses. Ultimately, the total number of events is dependent on the trajectory of the COVID-19 pandemic in Scotland, and statistical power will be assessed on an outcome-by-outcome basis.

### Patient and public involvement

An EAVE II Patient and public involvement (PPI) group has been established, which includes diverse ethnic and socioeconomic representation. Study plans have been discussed with the group and this has confirmed the importance of investigating a broad range of social factors (including SEP, as well as ethnicity). The study design was initially developed by the study investigators without PPI due to delays in being able to establish the PPI group, but more substantive input into the creation of detailed analysis plans is now ongoing. Findings will be discussed regularly with the group and revised on the basis of their input accordingly. The PPI group will also inform the development of dissemination plans.

### ETHICS AND DISSEMINATION

This study, as part of the wider EAVE II project, has been approved by the National Research Ethics Committee, South East Scotland 02. The study has also received approval from the Public Benefit and Privacy Panel for Health and Social Care (2021–0015), and approval from the Statistics Public Benefit and Privacy Panel (2021–0115). Findings from this study will be published in peer-reviewed journals and presented at international conferences. Analyses will be presented to policy-makers within UK and Scottish governments and to relevant public health practitioners in Scotland throughout the course of the study. Strengthening the Reporting of Observational Studies in Epidemiology and Reporting of studies Conducted using Observational Routinely-collected Data (RECORD) guidelines will be used in reporting the findings of the study.

**Author affiliations**
[1]MRC/CSO Social and Public Health Sciences Unit, University of Glasgow, Glasgow, UK
[2]Public Health Scotland, Edinburgh, UK
[3]Usher Institute, The University of Edinburgh, Edinburgh, UK
[4]Scottish Government, Edinburgh, UK
[5]Department of Mathematics and Statistics, University of Strathclyde, Glasgow, UK
[6]School of Medicine, University of St Andrews, St Andrews, UK
[7]General Practice and Primary Care, Aberdeen University, Aberdeen, UK
[8]Centre for Medical Informatics, The University of Edinburgh Usher Institute of Population Health Sciences and Informatics, Edinburgh, UK
[9]School of Medicine, University of St. Andrews, St. Andrews, UK
[10]Anaesthesia and Critical Care, University of Edinburgh, Edinburgh, UK
[11]Division of Community Health Sciences, University of Edinburgh, Edinburgh, UK
[12]Information Services Division, NHS National Services Scotland, Edinburgh, UK
[13]Wellington Faculty of Health, Victoria University of Wellington, Wellington, New Zealand

**Contributors** SVK designed the study. PH, EV, DB, KJH and CRS critically revised the study design. PH drafted the manuscript. EV, KJH, DB, EH, AHL, TA, CR, UA, LR, SJS, CM, AD, SK, JM, RW, EM, AS and SVK critically revised the manuscript. SVK obtained funding for the study.

**Funding** This study is being funded by the UKRI Centre on the Dynamics of Ethnicity 4 (ES/W000849/1). PH, SVK, AHL and KJH are funded by the Medical Research Council (MC_UU_00022/2) and Scottish Government Chief Scientist Office (SPHSU17). SVK is funded by a NRS Senior Clinical Fellowship (SCAF/15/02). EAVE II is funded by the Medical Research Council (MR/R008345/1) with the support of BREATHE-The Health Data Research Hub for Respiratory Health (MC_PC_19004), which is funded through the UK Research and Innovation Industrial Strategy Challenge Fund and delivered through Health Data Research UK; with additional support from the Scottish Government DG Health and Social Care.

**Competing interests** SVK is co-chair of the Scottish Government's Expert Reference Group on Ethnicity and COVID-19 and a member of the UK Government's Scientific Advisory Group on Emergencies (SAGE) subgroup on ethnicity and COVID-19. AS is a member of the Scottish Government's Chief Medical Officer's COVID-19 Advisory Group, the Scottish Government's Expert Reference Group on ethnicity and COVID-19 and the UK Government's Subgroup on Risk Stratification of the New and Emerging Respiratory Virus Threats Group (NERVTAG). LR acts as a medical advisor to Scottish Government on primary care and public health, including vaccination issues. CRS reports grants from the UK National Institute for Health Research, Medical Research Council and New Zealand Health Research Council and The Ministry of Business, Innovation and Employment.

**Patient and public involvement** Patients and/or the public were involved in the design, or conduct, or reporting, or dissemination plans of this research. Refer to the Methods section for further details.

**Patient consent for publication** Not required.

**Provenance and peer review** Not commissioned; externally peer reviewed.

**ORCID iDs**
Paul Henery http://orcid.org/0000-0003-0380-738X
Eleftheria Vasileiou http://orcid.org/0000-0001-6850-7578
Kirsten J Hainey http://orcid.org/0000-0002-6229-4556
Sarah Jane Stock http://orcid.org/0000-0003-4308-856X
Rachael Wood http://orcid.org/0000-0003-4453-623X
Srinivasa Vittal Katikireddi http://orcid.org/0000-0001-6593-9092

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
