## [Reviewer comments · BMJ Open]

ARTICLE DETAILS

TITLE (PROVISIONAL)	Ethnic and social inequalities in COVID-19 outcomes in Scotland: protocol for early pandemic evaluation and enhanced surveillance of COVID-19 (EAVE II)
AUTHORS	Henery, Paul; Vasileiou, Eleftheria; Hainey, Kirsten; Buchanan, Duncan; Harrison, Ewen; Leyland, Alastair; Alexis, Thomas; Robertson, Chris; Agrawal, Utkarsh; Ritchie, Lewis; Stock, Sarah; McCowan, Colin; Docherty, Annemarie; Kerr, Steven; Marple, James; Wood, Rachael; Moore, Emily; Simpson, Colin; Sheikh, Aziz; Katikireddi, Srinivasa

VERSION 1 – REVIEW

REVIEWER	Pecoraro, Valentina Nuovo Ospedale Civile Sant'Agostino Estense di Baggiovara
REVIEW RETURNED	09-Feb-2021
GENERAL COMMENTS	Authors submitted a study protocol aim to describe the epidemiology of COVID-19 in Scotland by social factors. This topic is interesting. I suggest to consider only one primary objective.

VERSION 1 – AUTHOR RESPONSE

Reviewer: 1

1. Authors submitted a study protocol aim to describe the epidemiology of COVID-19 in Scotland by social factors. This topic is interesting.

Thank you for your kind comments on our paper. We hope the changes below are to your satisfaction.

2. I suggest to consider only one primary objective.

We agree with your suggestion that we should focus on one primary objective. In light of this, we have reworded objective 1a as the singular primary objective of the study, and relegated 1b-e to secondary objectives (page 2, paragraph 2 and page 4, paragraphs 3-5).